# Deep Poisson Factor Modeling

**Ricardo Henao, Zhe Gan, James Lu and Lawrence Carin**
Department of Electrical and Computer Engineering
Duke University, Durham, NC 27708
`{r.henao,zhe.gan,james.lu,lcarin}@duke.edu`

## Abstract

We propose a new deep architecture for topic modeling, based on Poisson Factor Analysis (PFA) modules. The model is composed of a Poisson distribution to model observed vectors of counts, as well as a deep hierarchy of hidden binary units. Rather than using logistic functions to characterize the probability that a latent binary unit is on, we employ a Bernoulli-Poisson link, which allows PFA modules to be used repeatedly in the deep architecture. We also describe an approach to build discriminative topic models, by adapting PFA modules. We derive efficient inference via MCMC and stochastic variational methods, that scale with the number of non-zeros in the data and binary units, yielding significant efficiency, relative to models based on logistic links. Experiments on several corpora demonstrate the advantages of our model when compared to related deep models.

## 1 Introduction

Deep models, understood as multilayer modular networks, have been gaining significant interest from the machine learning community, in part because of their ability to obtain state-of-the-art performance in a wide variety of tasks. Their modular nature is another reason for their popularity. Commonly used modules include, but are not limited to, Restricted Boltzmann Machines (RBMs) [10], Sigmoid Belief Networks (SBNs) [22], convolutional networks [18], feedforward neural networks, and Dirichlet Processes[1] (DPs). Perhaps the two most well-known deep model architectures are the Deep Belief Network (DBN) [11] and the Deep Boltzmann Machine (DBM) [25], the former composed of RBM and SBN modules, whereas the latter is purely built using RBMs.

Deep models are often employed in topic modeling. Specifically, hierarchical tree-structured models have been widely studied over the last decade, often composed of DP modules. Examples of these include the nested Chinese Restaurant Process (nCRP) [1], the hierarchical DP (HDP) [27], and the nested HDP (nHDP) [23]. Alternatively, topic models built using modules other than DPs have been proposed recently, for instance the Replicated Softmax Model (RSM) [12] based on RBMs, the Neural Autoregressive Density Estimator (NADE) [17] based on neural networks, the Over-replicated Softmax Model (OSM) [26] based on DBMs, and Deep Poisson Factor Analysis (DPFA) [6] based on SBNs.

DP-based models have attractive characteristics from the standpoint of interpretability, in the sense that their generative mechanism is parameterized in terms of *distributions over topics*, with each topic characterized by a *distribution over words*. Alternatively, non-DP-based models, in which modules are parameterized by a deep hierarchy of *binary units* [12, 17, 26], do not have parameters that are as readily interpretable in terms of topics of this type, although model performance is often excellent. The DPFA model in [6] is one of the first representations that characterizes documents based on distributions over topics and words, while simultaneously employing a deep architecture based on binary units. Specifically, [6] integrates the capabilities of Poisson Factor Analysis (PFA)

Building upon the success of DPFA, this paper proposes a new deep architecture for topic modeling, based entirely on PFA modules. Our model fundamentally merges two key aspects of DP and non-DP-based architectures, namely: (*i*) its *fully nonnegative* formulation relies on Dirichlet distributions, and is thus readily interpretable throughout all its layers, not just at the base layer as in DPFA [6]; (*ii*) it adopts the rationale of traditional non-DP-based models such as DBNs and DBMs, by connecting layers via binary units, to enable learning of high-order statistics and structured correlations. The probability of a binary unit being on is controlled by a Bernoulli-Poisson link [30] (rather than a logistic link, as in the SBN), allowing repeated application of PFA modules at all layers of the deep architecture.

The main contributions of this paper are: (*i*) A deep architecture for topic models based entirely on PFA modules. (*ii*) Unlike DPFA, which is based on SBNs, our model has inherent shrinkage in all its layers, thanks to the DP-like formulation of PFA. (*iii*) DPFA requires sequential updates for its binary units, while in our formulation these are updated in block, greatly improving mixing. (*iv*) We show how PFA modules can be used to easily build discriminative topic models. (*v*) An efficient MCMC inference procedure is developed, that scales as a function of the number of *non-zeros* in the data and binary units. In contrast, models based on RBMs and SBNs scale with the size of the data and binary units. (*vi*) We also employ a scalable Bayesian inference algorithm based on the recently proposed Stochastic Variational Inference (SVI) framework [15].

## 2 Model

### 2.1 Poisson factor analysis as a module

We present the model in terms of document modeling and word counts, but the basic setup is applicable to other problems characterized by vectors of counts (and we consider such a non-document application when presenting results). Assume $\mathbf{x}_n$ is an $M$-dimensional vector containing word counts for the $n$-th of $N$ documents, where $M$ is the vocabulary size. We impose the model, $\mathbf{x}_n \sim \text{Poisson}\left(\mathbf{\Psi}(\boldsymbol{\theta}_n \circ \mathbf{h}_n)\right)$, where $\mathbf{\Psi} \in \mathbb{R}_+^{M \times K}$ is the factor loadings matrix with $K$ factors, $\boldsymbol{\theta}_n \in \mathbb{R}_+^K$ are factor intensities, $\mathbf{h}_n \in \{0,1\}^K$ is a vector of binary units indicating which factors are active for observation $n$, and $\circ$ represents the element-wise (Hadamard) product. One possible prior specification for this model, recently introduced in [32], is

$$x_{mn} = \sum_{k=1}^K x_{mkn}, \qquad x_{mkn} \sim \text{Poisson}(\lambda_{mkn}), \qquad \lambda_{mkn} = \psi_{mk}\theta_{kn}h_{kn},$$
$$\boldsymbol{\psi}_k \sim \text{Dirichlet}(\eta \mathbf{1}_M), \qquad \theta_{kn} \sim \text{Gamma}(r_k, (1-b)b^{-1}), \qquad h_{kn} \sim \text{Bernoulli}(\pi_{kn}), \tag{1}$$

where $\mathbf{1}_M$ is an $M$-dimensional vector of ones, and we have used the additive property of the Poisson distribution to decompose the $m$-th observed count of $\mathbf{x}_n$ as $K$ latent counts, $\{x_{mkn}\}_{k=1}^K$. Here, $\boldsymbol{\psi}_k$ is column $k$ of $\mathbf{\Psi}$, $x_{mn}$ is component $m$ of $\mathbf{x}_n$, $\theta_{kn}$ is component $k$ of $\boldsymbol{\theta}_n$, and $h_{kn}$ is component $k$ of $\mathbf{h}_n$. Furthermore, we let $\eta = 1/K$, $b = 0.5$ and $r_k \sim \text{Gamma}(1,1)$. Note that $\eta$ controls for the sparsity of $\mathbf{\Psi}$, while $r_k$ accommodates for over-dispersion in $\mathbf{x}_n$ via $\boldsymbol{\theta}_n$ (see [32] for details).

There is one parameter in (1) for which we have not specified a prior distribution, specifically $\mathbb{E}[p(h_{kn} = 1)] = \pi_{kn}$. In [32], $h_{kn}$ is provided with a beta-Bernoulli process prior by letting $\pi_{kn} = \pi_k \sim \text{Beta}(c\epsilon, c(1-\epsilon))$, meaning that every document has on average the same probability of seeing a particular topic as active, based on corpus-wide popularity. It further assumes topics are independent of each other. These two assumptions are restrictive because: (*i*) in practice, documents belong to a rather heterogeneous population, in which themes naturally occur within a corpus; letting documents have individual topic activation probabilities will allow the model to better accommodate for heterogeneity in the data. (*ii*) Some topics are likely to co-occur systematically, so being able to harness such correlation structures can improve the ability of the model for fitting the data.

The hierarchical model in (1), which in the following we denote as $\mathbf{x}_n \sim \text{PFA}(\mathbf{\Psi}, \boldsymbol{\theta}_n, \mathbf{h}_n; \eta, r_k, b)$, short for Poisson Factor Analysis (PFA), represents documents, $\mathbf{x}_n$, as purely additive combinations of up to $K$ topics (distributions over words), where $\mathbf{h}_n$ indicates what topics are active and $\boldsymbol{\theta}_n$, is the intensity of each one of the active topics that is manifested in document $\mathbf{x}_n$. It is also worth noting that the model in (1) is closely related to other widely known topic model approaches, such as Latent Dirichlet Allocation (LDA) [3], HDP [27] and Focused Topic Modeling (FTM) [29]. Connections between these models are discussed in Section 4.

## 2.2 Deep representations with PFA modules

Several models have been proposed recently to address the limitations described above [1, 2, 6, 27]. In particular, [6] proposed using multilayer SBNs [22], to impose correlation structure across topics, while providing each document with the ability to control its topic activation probabilities, without the need of a global beta-Bernoulli process [32]. Here we follow the same rationale as [6], but without SBNs. We start by noting that for a binary vector $\mathbf{h}_n$ with elements $h_{kn}$, we can write

$$h_{kn} = 1(z_{kn} \geq 1), \qquad z_{kn} \sim \text{Poisson}(\tilde{\lambda}_{kn}), \qquad (2)$$

where $z_{kn}$ is a latent count for variable $h_{kn}$, parameterized by a Poisson distribution with rate $\tilde{\lambda}_{kn}$; $1(\cdot) = 1$ if the argument is true, and $1(\cdot) = 0$ otherwise. The model in (2), recently proposed in [30], is known as the Bernoulli-Poisson Link (BPL) and is denoted $\mathbf{h}_n \sim \text{BPL}(\tilde{\boldsymbol{\lambda}}_n)$, for $\tilde{\boldsymbol{\lambda}}_n \in \mathbb{R}_+^K$. After marginalizing out the latent count $z_{kn}$ [30], the model in (2) has the interesting property that $p(h_{kn} = 1) = \text{Bernoulli}(\pi_{kn})$, where $\pi_{kn} = 1 - \exp(-\tilde{\lambda}_{kn})$. Hence, rather than using the logistic function to represent binary unit probabilities, we employ $\pi_{kn} = 1 - \exp(-\tilde{\lambda}_{kn})$.

In (1) and (2) we have represented the Poisson rates as $\lambda_{mkn}$ and $\tilde{\lambda}_{kn}$, respectively, to distinguish between the two. However, the fact that the count vector in (1) and the binary variable in (2) are both represented in terms of Poisson distributions suggests the following deep model, based on PFA modules (see graphical model in Supplementary Material):

$$\mathbf{x}_n \sim \text{PFA}\left(\boldsymbol{\Psi}^{(1)}, \boldsymbol{\theta}_n^{(1)}, \mathbf{h}_n^{(1)}; \eta^{(1)}, r_k^{(1)}, b^{(1)}\right), \qquad \mathbf{h}_n^{(1)} = \mathbf{1}\left(\mathbf{z}_n^{(2)}\right),$$

$$\mathbf{z}_n^{(2)} \sim \text{PFA}\left(\boldsymbol{\Psi}^{(2)}, \boldsymbol{\theta}_n^{(2)}, \mathbf{h}_n^{(2)}; \eta^{(2)}, r_k^{(2)}, b^{(2)}\right), \qquad \vdots$$

$$\vdots \qquad \mathbf{h}_n^{(L-1)} = \mathbf{1}\left(\mathbf{z}_n^{(L)}\right), \qquad (3)$$

$$\mathbf{z}_n^{(L)} \sim \text{PFA}\left(\boldsymbol{\Psi}^{(L)}, \boldsymbol{\theta}_n^{(L)}, \mathbf{h}_n^{(L)}; \eta^{(L)}, r_k^{(L)}, b^{(L)}\right), \qquad \mathbf{h}_n^{(L)} = \mathbf{1}\left(\mathbf{z}_n^{(L+1)}\right),$$

where $L$ is the number of layers in the model, and $\mathbf{1}(\cdot)$ is a vector operation in which each component imposes the left operation in (2). In this Deep Poisson Factor Model (DPFM), the binary units at layer $\ell \in \{1, \ldots, L\}$ are drawn $\mathbf{h}_n^{(\ell)} \sim \text{BPL}(\boldsymbol{\lambda}_n^{(\ell+1)})$, for $\boldsymbol{\lambda}_n^{(\ell)} = \boldsymbol{\Psi}^{(\ell)}(\boldsymbol{\theta}_n^{(\ell)} \circ \mathbf{h}_n^{(\ell)})$. The form of the model in (3) introduces latent variables $\{\mathbf{z}_n^{(\ell)}\}_{\ell=2}^{L+1}$ and the element-wise function $\mathbf{1}(\cdot)$, rather than explicitly drawing $\{\mathbf{h}_n^{(\ell)}\}_{\ell=1}^{L}$ from the BPL distribution. Concerning the top layer, we let $z_{kn}^{(L+1)} \sim \text{Poisson}(\lambda_k^{(L+1)})$ and $\lambda_k^{(L+1)} \sim \text{Gamma}(a_0, b_0)$.

## 2.3 Model interpretation

Consider layer 1 of (3), from which $\mathbf{x}_n$ is drawn. Assuming $\mathbf{h}_n^{(1)}$ is known, this corresponds to a focused topic model [29]. The columns of $\boldsymbol{\Psi}^{(1)}$ correspond to topics, with the $k$-th column $\boldsymbol{\psi}_k^{(1)}$ defining the probability with which words are manifested for topic $k$ (each $\boldsymbol{\psi}_k^{(1)}$ is drawn from a Dirichlet distribution, as in (1)). Generalizing the notation from (1), $\boldsymbol{\lambda}_{kn}^{(1)} = \boldsymbol{\psi}_k^{(1)} \theta_{kn}^{(1)} h_{kn}^{(1)} \in \mathbb{R}_+^M$ is the rate vector associated with topic $k$ and document $n$, and it is active when $h_{kn}^{(1)} = 1$. The word-count vector for document $n$ manifested from topic $k$ is $\mathbf{x}_{kn} \sim \text{Poisson}(\boldsymbol{\lambda}_{kn}^{(1)})$, and $\mathbf{x}_n = \sum_{k=1}^{K_1} \mathbf{x}_{kn}$, where $K_1$ is the number of topics in the model. The columns of $\boldsymbol{\Psi}^{(1)}$ define correlation among the words associated with the topics; for a given topic (column of $\boldsymbol{\Psi}^{(1)}$), some words co-occur with high probability, and other words are likely jointly absent.

We now consider a two-layer model, with $\mathbf{h}_n^{(2)}$ assumed known. To generate $\mathbf{h}_n^{(1)}$, we first draw $\mathbf{z}_n^{(2)}$, which, analogous to above, may be expressed as $\mathbf{z}_n^{(2)} = \sum_{k=1}^{K_2} \mathbf{z}_{kn}^{(2)}$, with $\mathbf{z}_{kn}^{(2)} \sim \text{Poisson}(\boldsymbol{\lambda}_{kn}^{(2)})$ and $\boldsymbol{\lambda}_{kn}^{(2)} = \boldsymbol{\psi}_k^{(2)} \theta_{kn}^{(2)} h_{kn}^{(2)}$. Column $k$ of $\boldsymbol{\Psi}^{(2)}$ corresponds to a meta-topic, with $\boldsymbol{\psi}_k^{(2)}$ a $K_1$-dimensional probability vector, denoting the probability with which each of the layer-1 topics are "on" when layer-2 "meta-topic" $k$ is on (i.e., when $h_{kn}^{(2)} = 1$). The columns of $\boldsymbol{\Psi}^{(2)}$ define correlation among the layer-1 topics; for a given layer-2 meta-topic (column of $\boldsymbol{\Psi}^{(2)}$), some layer-1 topics co-occur with high probability, and other layer-1 topics are likely jointly absent.

As one moves up the hierarchy, to layers $\ell > 2$, the meta-topics become increasingly more abstract and sophisticated, manifested in terms of probabilisitic combinations of topics and meta-topics at the layers below. Because of the properties of the Dirichlet distribution, each column of a particular $\boldsymbol{\Psi}^{(\ell)}$ is encouraged to be sparse, implying that a column of $\boldsymbol{\Psi}^{(\ell)}$ encourages use of a small subset of columns of $\boldsymbol{\Psi}^{(\ell-1)}$, with this repeated all the way down to the data layer, and the topics reflected in the columns of $\boldsymbol{\Psi}^{(1)}$. This deep architecture imposes correlation across the layer-1 topics, and it does it through use of PFA modules at all layers of the deep architecture, unlike [6] which uses an SBN for layers 2 through $L$, and a PFA at the bottom layer. In addition to the elegance of using a single class of modules at each layer, the proposed deep model has important computational benefits, as later discussed in Section 3.

## 2.4 PFA modules for discriminative tasks

Assume that there is a label $y_n \in \{1, \ldots, C\}$ associated with document $n$. We seek to learn the model for mapping $\mathbf{x}_n \rightarrow y_n$ simultaneously with learning the above deep topic representation. In fact, the mapping $\mathbf{x}_n \rightarrow y_n$ is based on the deep generative process for $\mathbf{x}_n$ in (3). We represent $y_n$ via the $C$-dimensional *one-hot* vector $\widehat{\mathbf{y}}_n$, which has all elements equal to zero except one, with the non-zero value (which is set to one) located at the position of the label. We impose the model

$$\widehat{\mathbf{y}}_n \sim \text{Multinomial}(1, \widehat{\boldsymbol{\lambda}}_n), \qquad \widehat{\lambda}_{cn} = \lambda_{cn} / \textstyle\sum_{c=1}^{C} \lambda_{cn}, \tag{4}$$

where $\widehat{\lambda}_{cn}$ is element $c$ of $\widehat{\boldsymbol{\lambda}}_n$, $\boldsymbol{\lambda}_n = \mathbf{B}(\boldsymbol{\theta}_n^{(1)} \circ \mathbf{h}_n^{(1)})$ and $\mathbf{B} \in \mathbb{R}_+^{C \times K}$, is a matrix of nonnegative classification weights, with prior distribution $\mathbf{b}_k \sim \text{Dirichlet}(\zeta \mathbf{1}_C)$, where $\mathbf{b}_k$ is a column of $\mathbf{B}$. Combining (3) with (4) allows us to learn the mapping $\mathbf{x}_n \rightarrow y_n$ via the shared first-layer local representation, $\boldsymbol{\theta}_n^{(1)} \circ \mathbf{h}_n^{(1)}$, that encodes topic usage for document $n$. This sharing mechanism allows the model to learn topics, $\boldsymbol{\Psi}^{(1)}$, and meta-topics, $\{\boldsymbol{\Psi}^{(\ell)}\}_{\ell=2}^{L}$, biased towards discrimination, as opposed to just explaining the data, $\mathbf{x}_n$. We call this construction *discriminative* deep Poisson factor modeling. It is worth noting that this is the first time that PFA and multi-class classification have been combined into a joint model. Although other DP-based discriminative topic models have been proposed [16, 21], they rely on approximations in order to combine the topic model, usually LDA, with softmax-based classification approaches.

# 3 Inference

A very convenient feature of the model in (3) is that all its conditional posterior distributions can be written in closed form due to local conjugacy. In this section, we focus on Markov chain Monte Carlo (MCMC) via Gibbs sampling as reference implementation and a stochastic variational inference approach for large datasets, where the fully Bayesian treatment becomes prohibitive.

Other alternatives for scaling up inference in Bayesian models such as the parameter server [13, 19], conditional density filtering [9] and stochastic gradient-based approaches [4, 5, 28] are left as interesting future work.

**MCMC** Due to local conjugacy, Gibbs sampling for the model in (3) amounts to sampling in sequence from the conditional posterior of all the parameters of the model, namely $\{\boldsymbol{\Psi}^{(\ell)}, \boldsymbol{\theta}_n^{(\ell)}, \mathbf{h}_n^{(\ell)}, r_k^{(\ell)}\}_{\ell=1}^{L}$ and $\boldsymbol{\lambda}^{(L+1)}$. The remaining parameters of the model are set to fixed values: $\eta = 1/K$, $b = 0.5$ and $a_0 = b_0 = 1$. We note that priors for $\eta$, $b$, $a_0$ and $b_0$ exist that result in Gibbs-style updates, and can be easily incorporated into the model if desired; however, we opted to keep the model as simple as possible, without compromising flexibility. The most unique conditional posteriors are shown below, without layer index for clarity,

$$\begin{aligned}
\boldsymbol{\psi}_k &\sim \text{Dirichlet}(\eta + x_{1k\cdot}, \ldots, \eta + x_{Mk\cdot}), \\
\theta_{kn} &\sim \text{Gamma}(r_k h_{kn} + x_{\cdot kn}, b^{-1}), \\
h_{kn} &\sim \delta(x_{\cdot kn} = 0)\text{Bernoulli}(\tilde{\pi}_{kn}(\tilde{\pi}_{kn} + 1 - \pi_{kn})^{-1}) + \delta(x_{\cdot kn} \geq 1),
\end{aligned} \tag{5}$$

where $x_{mk\cdot} = \sum_{n=1}^{N} x_{mkn}$, $x_{\cdot kn} = \sum_{m=1}^{M} x_{mkn}$ and $\tilde{\pi}_{kn} = \pi_{kn}(1-b)^{r_k}$. Omitted details, including those for the discriminative DPFM in Section 2.4, are given in the Supplementary Material.

Initialization is done at random from prior distributions, followed by layer-wise fitting (*pre-training*). In the experiments, we run 100 Gibbs sampling cycles per layer. In preliminary trials we observed that 50 cycles are usually enough to obtain good initial values of the global parameters of the model, namely $\{\boldsymbol{\Psi}^{(\ell)}, r_k^{(\ell)}\}_{\ell=1}^L$ and $\boldsymbol{\lambda}^{(L+1)}$.

**Stochastic variational inference (SVI)**  SVI is a scalable algorithm for approximating posterior distributions consisting of EM-style local-global updates, in which subsets of a dataset (*mini-batches*) are used to update in closed-form the variational parameters controlling both the local and global structure of the model in an iterative fashion [15]. This is done by using stochastic optimization with noisy natural gradients to optimize the variational objective function. Additional details and theoretical foundations of SVI can be found in [15].

In practice the algorithm proceeds as follows, where again we have omitted the layer index for clarity: ($i$) let $\{\boldsymbol{\Psi}^{(t)}, r_k^{(t)}, \boldsymbol{\lambda}^{(t)}\}$ be the global variables at iteration $t$. ($ii$) Sample a mini-batch from the full dataset. ($iii$) Compute updates for the variational parameters of the local variables using

$$\phi_{mkn} \propto \exp(\mathbb{E}[\log \psi_{mk}] + \mathbb{E}[\log \theta_{kn}]),$$

$$\theta_{kn} \sim \text{Gamma}(\mathbb{E}[r_k]\mathbb{E}[h_{kn}] + \textstyle\sum_{m=1}^M \phi_{mkn}, b^{-1}),$$

$$h_{kn} \sim \mathbb{E}[p(x_{\cdot kn}=0)]\text{Bernoulli}(\mathbb{E}[\tilde{\pi}_{kn}](\mathbb{E}[\tilde{\pi}_{kn}]+1-\mathbb{E}[\pi_{kn}])^{-1}) + \mathbb{E}[p(x_{\cdot kn}\geq 1)],$$

where $\mathbb{E}[x_{mkn}] = \phi_{mkn}$ and $\mathbb{E}[\tilde{\pi}_{kn}] = \mathbb{E}[\pi_{kn}](1-b)^{\mathbb{E}[r_k]}$. In practice, expectations for $\theta_{kn}$ and $h_{kn}$ are computed in log-domain. ($iv$) Compute a local update for the variational parameters of the global variables (only $\boldsymbol{\Psi}$ is shown) using

$$\widehat{\psi}_{mk} = \eta + N N_B^{-1} \textstyle\sum_{n=1}^{N_B} \phi_{mkn}, \tag{6}$$

where $N$ and $N_B$ are sizes of the corpus and mini-batch, respectively. Finally, we update the global variables as $\psi_k^{(t+1)} = (1-\rho_t)\psi_k^{(t)} + \rho_t\widehat{\psi}_k$, where $\rho_t = (t+\tau)^{-\kappa}$. The forgetting rate, $\kappa \in (0.5, 1]$ controls how fast previous information is forgotten and the delay, $\tau \geq 0$, down-weights early iterations. These conditions for $\kappa$ and $\tau$ guarantee that the iterative algorithm converges to a local optimum of the variational objective function. In the experiments, we set $\kappa = 0.7$ and $\tau = 128$. Additional details of the SVI algorithm for the model in (3) are given in the Supplementary Material.

**Importance of computations scaling as a function of number of non-zeros**  From a practical standpoint, the most important feature of the model in (3) is that inference does not scale as a function of the size of the corpus, but as a function of its number of non-zero elements, which is advantageous in cases where the input data is sparse (often the case). For instance, 2% of the entries in the widely studied 20 Newsgroup corpus are non-zero; similar proportions are also observed in the Reuters and Wikipedia data. Furthermore, this feature also extends to all the layers of the model regardless of $\{\mathbf{h}_n^{(\ell)}\}$ being latent. Similarly, for the discriminative DPFM in Section 2.4, inference scales with $N$, not $CN$, because the binary vector $\widehat{\mathbf{y}}_n$ has a single non-zero entry. This is particularly appealing in cases where $C$ is large.

In order to show that this scaling behavior holds, it is enough to see that by construction, from (1), if $x_{mn} = \sum_{k=1}^K x_{mkn} = 0$ (or $z_{mn}^{(\ell)}$ for $\ell > 1$), thus $x_{mkn} = 0$, $\forall k$ with probability 1. Besides, from (2) we see that if $h_{kn} = 0$ then $z_{kn} = 0$ with probability 1. As a result, update equations for all parameters of the model except for $\{\mathbf{h}_n^{(\ell)}\}$, depend only on non-zero elements of $\mathbf{x}_n$ and $\{\mathbf{z}_n^{(\ell)}\}$. Updates for the binary variables can be cheaply obtained in block from $h_{kn}^{(\ell)} \sim \text{Bernoulli}(\pi_{kn}^{(\ell)})$ via $\tilde{\lambda}_{kn}^{(\ell)}$, as previously described.

It is worth mentioning that models based on multinomial or Poisson likelihoods such as LDA [3], HDP [27], FTM [29] and PFA [32], also enjoy this property. However, the recently proposed deep PFA [6], does not use PFA modules on layers other than the first one. It uses SBNs or RBMs that are known to scale with the number of binary variables as opposed to their non-zero elements.

## 4  Related work

**Connections to other DP-based topic models**  PFA is a nonnegative matrix factorization model with Poisson link that is closely related to other DP-based models. Specifically, [32] showed that

by making $p(h_{kn} = 1) = 1$ and letting $\theta_{kn}$ have a Dirichlet, instead of a Gamma distribution as in (1), we can recover LDA by using the equivalence between Poisson and multinomial distributions. By looking at (5)-(6), we see that PFA and LDA have the same blocked Gibbs [3] and SVI [14] updates, respectively, when Dirichlet distributions for $\theta_{kn}$ are used. In [32], the authors showed that using the Poisson-gamma representation of the negative binomial distribution and a beta-Bernoulli specification for $p(h_{kn})$ in (1), we can recover the FTM formulation and inference in [29]. More recently, [31] showed that PFA is comparable to HDP in that the former builds group-specific DPs with normalized gamma processes. A more direct relationship between a three-layer HDP [27] and a two-layer version of (3) can be established by grouping documents by categories. In the HDP, three DPs are set for topics, document-wise topic usage and category-wise topic usage. In our model, $\mathbf{\Psi}^{(1)}$ represent $K_1$ topics, $\boldsymbol{\theta}_n^{(1)} \circ \mathbf{h}_n^{(1)}$ encodes document-wise topic usage and $\mathbf{\Psi}^{(2)}$ encodes topic usage for $K_2$ categories. In HDP, documents are assigned to categories *a priori*, but in our model document-category *soft* assignments are estimated and encoded via $\boldsymbol{\theta}_n^{(2)} \circ \mathbf{h}_n^{(2)}$. As a result, the model in (3) is a more flexible alternative to HDP in that it groups documents into categories in an unsupervised manner.

**Similar models**  Non-DP-based deep models for topic modeling employed in the deep learning literature typically utilize RBMs or SBNs as building blocks. For instance, [12] and [20] extended RBMs via DBNs to topic modeling and [26] proposed the over-replicated softmax model, a deep version of RSM that generalizes RBMs.

Recently, [24] proposed a framework for generative deep models using exponential family modules. Although they consider Poisson-Poisson and Gamma-Gamma factorization modules akin to our PFA modules, their model lacks the explicit binary unit linking between layers commonly found in traditional deep models. Besides, their inference approach, *black-box* variational inference, is not as conceptually simple, but it scales with the number of non-zeros as our model.

DPFA, proposed in [6], is the model closest to ours. Nevertheless, our proposed model has a number of key differentiating features. ($i$) Both of them learn topic correlations by building a multilayer modular representation on top of PFA. Our model uses PFA modules throughout all layers in a conceptually simple and easy to interpret way. DPFA uses Gaussian distributed weight matrices within SBN modules; these are hard to interpret in the context of topic modeling. ($ii$) SBN architectures have the shortcoming of not having block closed-form conditional posteriors for their binary variables, making them difficult to estimate, especially as the number of variables increases. ($iii$) Factor loading matrices in PFAs have natural shrinkage to counter overfitting, thanks to the Dirichlet prior used for their columns. In SBN-based models, shrinkage has to be added via variable augmentation at the cost of increasing inference complexity. ($iv$) Inference for SBN modules scales with the number of hidden variables in the model, not with the number of non-zero elements, as in our case.

## 5   Experiments

**Benchmark corpora**  We present experiments on three corpora: 20 Newsgroups (20 News), Reuters corpus volume I (RCV1) and Wikipedia (Wiki). 20 News is composed of 18,845 documents and 2,000 words, partitioned into a 11,315 training set and a 7,531 test set. RCV1 has 804,414 newswire articles containing 10,000 words. A random 10,000 subset of documents is used for testing. For Wiki, we obtained $10^7$ random documents, from which a subset of 1,000 is set aside for testing. Following [14], we keep a vocabulary consisting of 7,702 words taken from the top 10,000 words in the Project Gutenberg Library.

As performance measure we use held-out perplexity, defined as the geometric mean of the inverse marginal likelihood of every word in the set. We cannot evaluate the intractable marginal for our model, thus we compute the *predictive perplexity* on a 20% subset of the held-out set. The remaining 80% is used to learn document-specific variables of the model. The training set is used to estimate the global parameters of the model. Further details on perplexity evaluation for PFA models can be found in [6, 32].

We compare our model (denoted DPFM) against LDA [3], FTM [29], RSM [12], nHDP [23] and DPFA with SBNs (DPFA-SBN) and RBMs (DPFA-RBM) [6]. For all these models we use the settings described in [6]. Inference methods for RSM and DPFA are contrastive divergence with

Table 1: Held-out perplexities for 20 News, RCV1 and Wiki. Size indicates number of topics and/or binary units, accordingly.

| Model | Method | Size | 20 News | RCV1 | Wiki |
|---|---|---|---|---|---|
| DPFM | SVI | 128-64 | 818 | 961 | 791 |
| DPFM | MCMC | 128-64 | **780** | **908** | 783 |
| DPFA-SBN | SGNHT | 1024-512-256 | —— | 942 | **770** |
| DPFA-SBN | SGNHT | 128-64-32 | 827 | 1143 | 876 |
| DPFA-RBM | SGNHT | 128-64-32 | 896 | 920 | 942 |
| nHDP | SVI | (10,10,5) | 889 | 1041 | 932 |
| LDA | Gibbs | 128 | 893 | 1179 | 1059 |
| FTM | Gibbs | 128 | 887 | 1155 | 991 |
| RSM | CD5 | 128 | 877 | 1171 | 1001 |

step size 5 (CD5) and stochastic gradient Nse-Hoover thermostats (SGNHT) [5], respectively. For our model, we run 3,000 samples (first 2,000 as burnin) for MCMC and 4,000 iterations with 200-document mini-batches for SVI. For the Wiki corpus, MCMC-based DPFM is run on a random subset of $10^6$ documents. The code used, implemented in Matlab, will be made publicly available.

Table 1 show results for the corpora being considered. Figures for methods other than DPFM were taken from [6]. We see that multilayer models (DPFM, DPFA and nHDP) consistently outperform single layer ones (LDA, FTM and RSM), and that DPFM has the best performance across all corpora for models of comparable size. OSM result (not shown) are about 20 units better than RSM in 20 News and RCV1, see [26]. We also see that MCMC yields better perplexities when compared to SVI. The difference in performance between these two inference methods is likely due to the mean-field approximation and the online nature of SVI. We verified empirically (results not shown) that doubling the number of hidden units, adding a third layer or increasing the number of samples/iterations for DPFM does not significantly change the results in Table 1. As a note on computational complexity, one iteration of the two-layer model on the 20 News corpus takes approximately 3 and 2 seconds, for MCMC and SVI, respectively. For comparison, we also ran the DPFA-SBN model in [6] using a two-layer model of the same size; in their case it takes about 24, 4 and 5 seconds to run one iteration using MCMC, conditional density filtering (CDF) and SGNHT, respectively. Runtimes for DPFA-RBM are similar to those of DPFA-SBN, LDA and RSM are faster than 1-layer DPFM, FTM is comparable to the latter, and nHDP is slower than DPFM.

Figure 1 shows a representative meta-topic, $\psi_k^{(2)}$, from the two-layer model for 20 News. For the five largest weights in $\psi_k^{(2)}$ (y-axis), which correspond to layer-1 topic indices (x-axis), we also show the top five words in their layer-1 topic, $\psi_k^{(1)}$. We observe that this meta-topic is loaded with religion specific topics, judging by the words in them. Additional graphs, and tables showing the top words in each topic for 20 News and RCV1 are provided in the Supplementary Material.

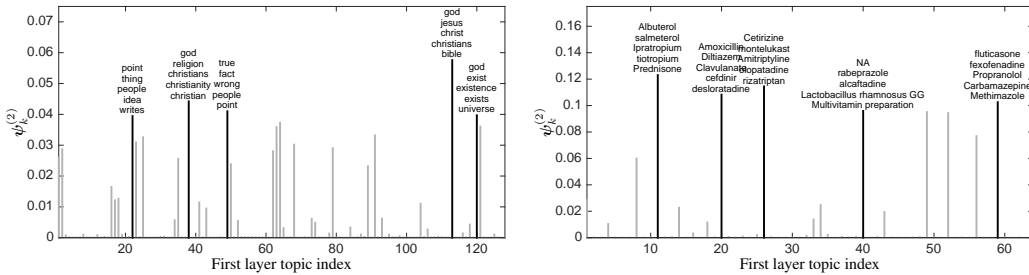

Figure 1: Representative meta-topics obtained from (left) 20 News and (right) medical records. Meta-topic weights $\psi_k^{(2)}$ vs. layer-1 topics indices, with word lists corresponding to the top five words in layer-1 topics, $\psi_k^{(1)}$.

**Classification**    We use 20 News for document classification, to evaluate the discriminative DPFM model described in Section 2.4. We use test set accuracy on the 20-class task as performance measure and compare our model against LDA, DocNADE [17], RSM and OSM. Results for these four models were obtained from [26], where multinomial logistic regression with cross-entropy loss func-

Table 2: Test accuracy on 20 News. Subscript accompanying model names indicate their size.

| Model | $LDA_{128}$ | $DocNADE_{512}$ | $RSM_{512}$ | $OSM_{512}$ | $DPFM_{128}$ | $DPFM_{128-64}$ |
|---|---|---|---|---|---|---|
| Accuracy (%) | 65.7 | 68.4 | 67.7 | 69.1 | 72.11 | **72.67** |

tion was used as classification module. Test accuracies in Table 2 show that our model significantly outperforms the others being considered. Note as well that our one-layer model still improves upon the four times larger OSM, by more than 3%. We verified that our two-layer model outperforms well known supervised methods like multinomial logistic regression, SVM, supervised LDA and two-layer feedforward neural networks, for which test accuracies ranged from 67% to 72.14%, using term frequency-inverse document frequency features. We could not improve results by increasing the size of our model, however, we may be able to do so by following the approach of [33], where a single classification module (SVM) is shared by 20 one-layer topic models (LDAs). Exploration of more sophisticated deep model architectures for discriminative DPFMs is left as future work.

**Medical records**  The Duke University Health System medical records database used here, is a 5 year dataset generated within a large health system including three hospitals and an extensive network of outpatient clinics. For this analysis, we utilized self-reported medication usage from over 240,000 patients that had over 4.4 million patient visits. These patients reported over 34,000 different types of medications which were then mapped to one of 1,691 pharmaceutical active ingredients (AI) taken from RxNorm, a depository of medication information maintained by the National Library of Medicine that includes trade names, brand names, dosage information and active ingredients. Counts for patient-medication usage reflected the number of times an AI appears in a patients record. Compound medications that include multiple active ingredients incremented counts for all AI in that medication. Removing AIs with less than 10 overall occurrences and patients lacking medication information, results in a 1,019×131,264 matrix of AIs vs. patients.

Results for a MCMC-based DPFM of size 64-32, with the same setting used for the first experiment, indicate that *pharmaceutical topics* derived from this analysis form clinically reasonable clusters of pharmaceuticals, that may be prescribed to patients for various ailments. In particular, we found that layer-1 topic 46 includes a cluster of insulin products: Insulin Glargine, Insulin Lispro, Insulin Aspart, NPH Insulin and Regular Insulin. Insulin dependent type-2 diabetes patients often rely on tailored mixtures of insulin products with different pharmacokinetic profiles to ensure glycemic control. In another example, we found in layer-1 topic 22, an Angiotensin Receptor Blocker (ARB), Losartan with a HMGCoA Reductase inhibitor, Atorvastatin and a heart specific beta blocker, Carvedilol. This combination of medications is commonly used to control hypertension and hyperlipidemia in patients with cardiovascular risk. The second layer correlation structure between topics of drug products also provide interesting composites of patient types based on the first-layer pharmaceutical topics. Specifically, layer-2 factor 22 in Figure 1 reveals correlation between layer-1 drug factors that would be used to treat types of respiratory patients that had chronic obstructive respiratory disease and/or asthma (Albuterol, Montelukast) and seasonal allergies. Additional graphs, including top medications for all pharmaceutical topics found by our model are provided in the Supplementary Material.

## 6  Conclusion

We presented a new deep model for topic modeling based on PFA modules. We have combined the interpretability of DP-based specifications found in traditional topic models with deep hierarchies of hidden binary units. Our model is elegant in that a single class of modules is used at each layer, but at the same time, enjoys the computational benefit of scaling as a function of the number of zeros in the data and binary units. We described a discriminative extension for our deep architecture, and two inference methods: MCMC and SVI, the latter for large datasets. Compelling experimental results on several corpora and on a new medical records database demonstrated the advantages of our model.

Future directions include working towards alternatives for scaling up inference algorithms based on gradient-based approaches, extending the use of PFA modules in deep architectures to more sophisticated discriminative models, multi-modal tasks with mixed data types, and time series modeling using ideas similar to [8].

**Acknowledgements**  This research was supported in part by ARO, DARPA, DOE, NGA and ONR.

## Footnotes

[1]Deep models based on DP priors are usually called *hierarchical models*.

[32] with a deep architecture composed of SBNs [7]. PFA is a nonnegative matrix factorization framework closely related to DP-based models. Results in [6] show that DPFA outperforms other well-known deep topic models.

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
