[Supplementary Material]

# Deep Poisson Factor Modeling

**Ricardo Henao, Zhe Gan, James Lu and Lawrence Carin**
Department of Electrical and Computer Engineering
Duke University, Durham, NC 27708
{r.henao,zhe.gan,james.lu,lcarin}@duke.edu

## 1 Graphical Model

Figure 1: Graphical models. (a) Poisson Factor Analysis (PFA) module. Nodes ($\mathbf{b}_k$, $\hat{\mathbf{z}}_n$ and $\mathbf{y}_n$) and edges drawn with dashed lines correspond to the discriminative PFA. (b) Deep Poisson factor model. Filled and empty nodes represent observed and latent variables, respectively.

## 2 Inference Details

### 2.1 MCMC

Conditional posteriors (layer index omitted for clarity):

$$\boldsymbol{\psi}_k \sim \text{Dirichlet}(\eta + x_{1k\cdot}, \ldots, \eta + x_{Mk\cdot}),$$

$$\theta_{kn} \sim \text{Gamma}(r_k h_{kn} + x_{\cdot kn}, b^{-1}),$$

$$h_{kn} \sim \delta(x_{\cdot kn} = 0)\text{Bernoulli}(\tilde{\pi}_{kn}(\tilde{\pi}_{kn} + 1 - \pi_{kn})^{-1}) + \delta(x_{\cdot kn} \geq 1),$$

$$r_k \sim \text{Gamma}\left(1 + \sum_n u_{kn}, 1 - \sum_n h_{kn}\log(1-b)\right),$$

$$z_{kn} \sim \delta(h_{kn} = 1)\text{Poisson}_+(\tilde{\lambda}_{kn}),$$

where $\mathrm{Poisson}_+(\cdot)$ is the zero-truncated Poisson distribution and

$$x_{mk\cdot} = \sum_{n=1}^{N} x_{mkn}\,,$$

$$x_{\cdot kn} = \sum_{m=1}^{M} x_{mkn}\,,$$

$$\tilde{\pi}_{kn} = \pi_{kn}(1-b)^{r_k}\,,$$

$$u_{kn} = \sum_{j=1}^{x_{\cdot kn}} u_{knj}\,, \qquad u_{knj} \sim \mathrm{Bernoulli}\left(\frac{r_k}{r_k + j - 1}\right)\,.$$

(1)

Note that for multilayer models, $\pi_{kn}^{(\ell)} = 1 - \exp(\lambda_{kn}^{(\ell+1)})$. The data augmentation scheme for $r_k$ via $u_{kn}$ is described in [1].

For the discriminative DPFA, lets denote latent counts for $\widehat{y}_n$ as $\widehat{x}_{ckn}$, with summaries analogous to (1), as $\widehat{x}_{ck\cdot}$ and $\widehat{x}_{\cdot kn}$. Then,

$$\mathbf{b}_k \sim \mathrm{Dirichlet}(\zeta + \widehat{x}_{1k\cdot}, \ldots, \zeta + \widehat{x}_{Ck\cdot})\,,$$

$$\theta_{kn} \sim \mathrm{Gamma}(r_k h_{kn} + x_{\cdot kn} + \widehat{x}_{\cdot kn}, b^{-1})\,,$$

$$h_{kn} \sim \delta(x_{\cdot kn} = 0 \wedge \widehat{x}_{\cdot kn} = 0)\mathrm{Bernoulli}(\tilde{\pi}_{kn}(\tilde{\pi}_{kn} + 1 - \pi_{kn})^{-1}) + \delta(x_{\cdot kn} \geq 1 \vee \widehat{x}_{\cdot kn} \geq 1)\,.$$

Provided that $\boldsymbol{\theta}_n$ and $\mathbf{h}_n$ are shared by two PFAs, one for the count data, $\mathbf{x}_n$, and the other for the labels, $\widehat{y}_n$, their conditional posteriors are functions of latent counts coming from both sources, $x_{\cdot kn}$ and $\widehat{x}_{\cdot kn}$, respectively.

## 2.2 SVI

Variational parameter updates using (layer index omitted for clarity):

$$\phi_{mkn} \propto \exp(\mathbb{E}[\log \psi_{mk}] + \mathbb{E}[\log \theta_{kn}])\,,$$

$$\theta_{kn} \sim \mathrm{Gamma}(\mathbb{E}[r_k]\mathbb{E}[h_{kn}] + \sum_{m=1}^{M} \phi_{mkn}, b^{-1})\,,$$

$$h_{kn} \sim \mathbb{E}[p(x_{\cdot kn} = 0)]\mathrm{Bernoulli}(\mathbb{E}[\tilde{\pi}_{kn}](\mathbb{E}[\tilde{\pi}_{kn}] + 1 - \mathbb{E}[\pi_{kn}])^{-1}) + \mathbb{E}[p(x_{\cdot kn} \geq 1)]\,,$$

$$r_k \sim \mathrm{Gamma}\left(1 + \sum_n \mathbb{E}[u_{kn}], 1 - \sum_n \mathbb{E}[p(h_{kn} = 1)]\log(1-b)\right)\,,$$

$$z_{kn} \sim \mathbb{E}[p(h_{kn} = 1)]\mathrm{Poisson}_+(\tilde{\lambda}_{kn})\,,$$

where $\mathbb{E}[x_{mkn}] = \phi_{mkn}$, $\mathbb{E}[\tilde{\pi}_{kn}] = \mathbb{E}[\pi_{kn}](1-b)^{\mathbb{E}[r_k]}$ and $\mathbb{E}[u_{kn}] = \sum_{j=1}^{x_{\cdot kn}} \mathbb{E}[r_k](\mathbb{E}[r_k] + j - 1)^{-1}$.

## List of Figures

Figure 2: Representative meta-topics obtained from 20 News. Meta-topic weights $\psi_k^{(2)}$ vs. layer-1 topics indices, with word lists corresponding to the top four words in layer-1 topics, $\psi_k^{(1)}$.

Figure 3: Graph representation obtained from 20 News. Meta-topics are denoted by circles and layer-1 topics as boxes, with word lists corresponding to the top four words in layer-1 topics, $\psi_k^{(1)}$. For clarity, we only show the top four connections between meta-topics and their associated topics

Figure 4: Graph representation obtained from RCV1. Meta-topics are denoted by circles and layer-1 topics as boxes, with word lists corresponding to the top four words in layer-1 topics, $\psi_k^{(1)}$. For clarity, we only show the top four connections between meta-topics and their associated topics.

Figure 5: Graph representation obtained from Wiki. Meta-topics are denoted by circles and layer-1 topics as boxes, with word lists corresponding to the top four words in layer-1 topics, $\psi_k^{(1)}$. For clarity, we only show the top four connections between meta-topics and their associated topics.

Figure 6: Representative meta-topics obtained from medical records data. Meta-topic weights $\psi_k^{(2)}$ vs. layer-1 topics indices, with word lists corresponding to the top four words in layer-1 topics, $\psi_k^{(1)}$.

Figure 7: Graph representation obtained from medical records data. Meta-topics are denoted by circles and layer-1 topics as boxes, with word lists corresponding to the top four words in layer-1 topics, $\psi_k^{(1)}$. For clarity, we only show the top four connections between meta-topics and their associated topics.

# References

[1] M. Zhou and L. Carin. Negative binomial process count and mixture modeling. *PAMI*, 2015.