[Reviews · NeurIPS 2015]

Submitted by Assigned_Reviewer_1

This paper proposes a novel probabilistic deep architecture for modelling count data. In addition the authors propose a modelling extension for multi-class classification problems (i.e., discriminative modelling). The authors derive two inference techniques, one sampling one and a variational inference one, and show empirical gains compared to unsupervised methods using several large-scale datasets.

Questions/Comments: - I found the model to be clearly explained and motivated. I enjoyed

reading Section 2.3. I think adding a figure to illustrate Equations 3

would be useful. - I think it may be slightly misleading to imply that [23] does not scale

wrt to the number of zeros. My understanding is that using a Poisson

likelihood allows all methods to effectively ignore the zeros. Further

it seems like most of the computational gains come from that rather than

from higher-layers (which are typically much smaller). [23] also does allow

non-linear activation functions. - Computation. It appears that your model scales well. It would be

interesting to have an idea of how computation scales and how long it

takes to learn on these larger datasets. Providing a rough comparison to

competing models (docNADE, LDA, and replicated softmax) would also be

useful. - When reporting the results of the classification experiments it seems

like you are only comparing to unsupervised techniques. In that sense

the comparison is not absolutely fair. It would be good to add, at

least, one simple supervised baseline (e.g., a small neural net with a

softmax output and the word frequencies as inputs).

Other comments: - line 165: Sparsity of the Dirichlet relies on your choice of parameters

(eta). I think it would be good to make it clear.
Summary: - This is a good paper. The model scales, and pushes our

understanding of deep generative models. The discriminative extension is

also worth noting. Empirical results are relatively good as well.

Submitted by Assigned_Reviewer_2

The paper presents a multilayer model of count vectors using Poisson Factor Analysis at all layers (providing interpretable topics) and binary units connecting these layers (learning topic correlations). MCMC and SVI inference is straightforward -- all conditional posteriors are in closed form --

and inference scales in the number of non-zero observations and hidden units. The model is an incremental change from Zhou et al. (2012), removing the gobal beta-Bernoulli process and using the Bernoulli-Poisson link to avoid using sigmoid belief networks.

Both the model and inference described in the paper are elegant. The model is only an incremental improvement from prior work (notably Zhou et al. (2012)), but it's likely to be of significant interest to the community.

1. The experimental analysis has been done with an eye towards comparing to other deep and one-layer models. However, there's hardly any effort in exploring the proposed model itself.

a. how did you fix the layer widths? have you studied more than 2 layers?

b. how does the model deal with overdispersion in data?

c. how should the layer widths decay with depth?

2. it's surprising that even your one-layer model does significantly better than ORSM and LDA (which is similar to PFA). is this due to your approach to discriminative topic models? there is no explanation provided.

3. why is ORSM not included in Table 1?
Summary: The paper presents a tweak to existing deep network topic models (combining ideas from Zhou et al. (2012) and Zhou et al. (2015)) and shows how a hierarchy of Poisson Factor Analysis units can be connected using hidden binary units.

Although an incremental contribution, both MCMC and variational inference are made much simpler due to local conjugacy and experimental results show superior performance. The paper is lacking in a experimental investigation into the network structure -- instead it seems to be arbitrarily fixed. It is still likely to be of much interest to the deep learning and topic modeling community.

Submitted by Assigned_Reviewer_3

The paper extends previous work on using deep Poisson Factor Analysis (PFA) for topic modeling by using a Bernoulli-Poisson link, instead of logistic functions. The paper also describes a way to jointly model documents and their associated discrete labels. Experiments show the proposed method outperform related baselines in held-out perplexities and classification accuracy.

The paper extends existing work (Gan et al, ICML'15 in particular) to provide a more flexible way to define the prior on documents' proportions over topics. Even though mostly a combination of existing ideas, I think the paper provides some advances in applying deep models for topic modeling. Here are some of my detailed comments:

- It is interesting to see that both MCMC and variational inference techniques are included. One of the arguments for using VI in the paper is its scalability. It would be interesting to see comparison on running time between the two inference techniques.

- Jointly capturing documents with their associated metadata is a well-studied problem. I am wondering why the performance of traditional supervised topic models such as sLDA (for classification) are not included for comparison in Section 5.

- For readers who are not familiar with conventional notations of deep models, I would suggest including some figure to illustrate the different layers and their input/output in Section 2.
Summary: The paper extends and improves previous work on using deep Poisson Factor Analysis for topic modeling. While mostly a combination of existing ideas, I think the paper provides some advances in applying deep models for topic modeling.

Author Feedback
Author rebuttal: We thank the reviewers for their encouragement and constructive criticism, we address their main concerns and comments below.

We use the following abbreviations: R1 refers to reviewer 1 and R1/3 refers to both reviewer 1 and 3.

R1/3/4/5: We will provide runtimes as proxies for the computational cost in the experiments.

R1/3/4: We will add results with supervised baselines: neural network and sLDA, as suggested by the reviewers.

R1/3: We did not include a graphical model in Section 2 due to space limitations but we will be happy to add it to the supplementary material.

R1: We will make clear that [23], as it's based on a Poisson likelihood allows to effectively ignore zeros and that it also allows for non-linear activation functions.

R1: We will make clearer that sparsity of the Dirichlet prior relies on the choice of \eta.

R2: We fixed layer widths to be consistent with DPFA's experiments. As stated in the paper, we also tried models with more hidden units per layer and one additional layer but the results were not significantly different from those shown in Table 1. It is worth noting that because our model has sparse loadings at all its layers, due to the Dirichlet prior used, the model should be fairly robust to layer width settings, as long as widths are large enough.

R2: The model accommodates well for overdispersion via r_k in the prior distribution for \theta, in fact, it can be shown that marginalizing out \theta in our Poisson-Gamma formulation reduces to a negative binomial likelihood with parameter r_k [30], meaning that our model effectively decouples mean and variance of observed data. This interesting point will be added to the paper.

R2: We did not investigate how layer widths should decay with depth but we agree with the reviewer that it would be interesting to see, we leave it as future work.

R2: We believe that the difference in performance between LDA, ORSM and our discriminative PFA is due to our joint learning approach, we will make it clearer.

R2: We will include ORSM results in Table 1, the difference in perplexity between RSM and ORSM is not that large, about 20 units in [25].

R4: We did not include DPFA in the classification results because the model was not originally conceived for that purpose [6]. We agree that a discriminative version of DPFA is possible but out of reach for our paper.